

# Potential of needle trap microextraction – portable gas chromatography – mass spectrometry for measurement of atmospheric volatile compounds

Luís Miguel Feijó Barreira[1], Yu Xue[1], Geoffroy Duporté[1], Jevgeni Parshintsev[1], Kari Hartonen[1], Matti Jussila[1], Markku Kulmala[2], Marja-Liisa Riekkola[1]

[1]Laboratory of Analytical Chemistry, Department of Chemistry, University of Helsinki, Helsinki, P.O. Box 55, FI-00014, Finland
[2]Division of Atmospheric Sciences, Department of Physics, University of Helsinki, Helsinki, P.O. Box 64, FI-00014, Finland

*Correspondence to*: M.-L. Riekkola (marja-liisa.riekkola@helsinki.fi)

**Abstract.** Volatile organic compounds (VOCs) play a key role in atmospheric chemistry and physics. They participate in photochemical reactions in the atmosphere, which have direct implications on climate through, e.g., aerosol particle formation. Forests are important sources of VOCs, and the limited resources and infrastructures often found in many remote environments call for the development of portable devices. In this research, the potential of needle trap microextraction and portable gas chromatography-mass spectrometry for the study of VOCs at forest site was evaluated. Measurements were performed in summer and autumn 2014 at the Station for Measuring Ecosystem-Atmosphere Relations (SMEAR II) in Hyytiälä, Finland. During the first part of the campaign (summer) the applicability of the developed method was tested for the determination of monoterpenes, pinonaldehyde, aldehydes, amines and anthropogenic compounds. The temporal variation of aerosol precursors was determined, and evaluated against temperature and aerosol number concentration data. The most abundant monoterpenes, pinonaldehyde and aldehydes were successfully measured, their relative amounts being lower during days when particle number concentration was higher. Ethylbenzene, para- and meta-xylene were also found when wind direction was from cities with substantial anthropogenic activity. An accumulation of VOCs in the snow cover was observed in the autumn campaign. Results demonstrated the successful applicability of needle trap microextraction and portable gas chromatography-mass spectrometry for the rapid *in-situ* determination of organic gaseous compounds in the atmosphere.

## 1 Introduction

Earth's atmosphere contains at trace levels a large variety of different volatile compounds that are produced from biogenic and anthropogenic sources. These trace gases can influence air quality and climate through the greenhouse effect (e.g. $CO_2$ and $CH_4$) and participation in atmospheric aerosol nucleation and growth (Kulmala, 2003).

Aerosols have a significant impact on climate, both directly, by scattering and absorbing radiation, and indirectly by acting as a cloud condensation nuclei (CCN) and ice nuclei (IN) that naturally affect cloud properties (Zelenyuk et al., 2010). Particles are classified as primary aerosols, when they are directly emitted into the atmosphere, and secondary organic aerosols (SOA),



when they are formed from gaseous biogenic or anthropogenic precursors in the atmosphere (Reinhardt et al., 2007). Secondary aerosol formation is recognized as the main source of aerosol particles in the global atmosphere, and organic vapors play a key role on their nucleation and growth mechanisms (Kulmala et al., 2013).

Many organic compounds found in the atmosphere, of both anthropogenic and biogenic origin, have been found to produce
SOA (e.g. Seinfeld and Pankow, 2003). The role of VOCs as a source of natural aerosols has been thoroughly studied in the background regions of the northern hemisphere. Coniferous forests produce large amounts of volatile vapors, especially terpenes (Bäck et al., 2012). In temperate and boreal regions, monoterpenes dominate all biogenic VOC (BVOC) emissions, α-pinene and $\Delta^3$-carene being the most emitted species (Rinne et al., 2009; Yassaa et al., 2012). Monoterpene emissions are mainly dependent on temperature, although the influence of solar radiation and reaction to stress including physical damage,
herbivoral attack, drought and heat have also been reported (Nölscher et al., 2012). Monoterpenes are oxidized in the atmosphere, and the produced oxygenated VOCs contribute to new particle formation and growth (e.g. Hao et al., 2009; Kourtchev et al., 2008; Laaksonen et al., 2008). Carbonyl compounds, such as aldehydes, are also important to atmospheric chemistry due to their atmospheric photolysis, reaction with hydroxyl and nitrate radicals, and their contribution to new particle formation (Atkinson, 2000; Jang and Kamens, 2001). Aldehydes can be directly emitted to the atmosphere from incomplete
combustion of biomass and fossil fuels, by vegetation, when exposed to stress conditions such as ozone exposure or insect and pathogen attacks, and formed in the atmosphere as a result of photochemical oxidation of reactive compounds (Altemose et al., 2015; Wildt et al., 2003). Primary emissions of saturated $C_6$-$C_{10}$, such as hexanal, heptanal, octanal, nonanal and decanal, were particularly found from several plant species (Wildt et al., 2003). The photo-oxidation products of benzene, toluene, ethylbenzene and m-,p-and o-xylene (BTEX) are compounds that are recognized as important precursors of SOA (Li et al.,
2014). BTEX sources are very diverse but vehicular exhausts and industrial emissions are considered the major ambient sources (e.g. Scheff and Wadden, 1993). An enhanced stabilization and nucleation rate by atmospheric basic compounds has been confirmed at given sulfuric acid and water concentrations (e.g. Chen et al., 2012; Kurtén et al., 2008; Yu et al., 2012). For example, alkylamines were found to participate in formation and growth of sub-3-nm particles (Yu et al., 2012). These compounds are emitted both from anthropogenic and biogenic sources, but due to their limited lifetimes, they are not
transported long distances in the atmosphere. In forest atmosphere, amine concentrations seem to be linked with vegetation, soil activity, and litterfall (Kieloaho et al., 2013).

Sampling of VOCs is mainly performed by using cartridges filled with an adsorbent material consisting usually of Tenax TA/Carbopack-B that are subsequently analysed by thermodesorption gas chromatography-mass spectrometry (e.g. Haapanala et al., 2012). Sampling of amines is typically carried out by using impingers, sorbent tubes, acid-coated denuders or solid-
phase microextraction (SPME), while analysis is normally performed by gas chromatography (GC), high-performance liquid chromatography (HPLC) or ion chromatography (IC) all coupled with mass spectrometry (MS) (Parshintsev et al., 2015; Szulejko and Kim, 2014). Most of these techniques require an additional extraction step or/and the use of a derivatization reagent. In addition, the reactivity and adsorption effects associated with small amines are the challenging factors during all analytical steps. VOCs can also be sampled by needle trap microextraction (NTME) that is considered to be more robust than



other extraction techniques such as solid-phase microextraction, since the sorbent particles are protected inside the needle. Furthermore, because NTME is an exhaustive technique, the sensitivity can be improved by increasing the sample volume; and capacity can be increased by increasing the volume of the sorbent packed in the needle (Eom et al., 2008). For fast *in-situ* analysis, portable gas chromatography-mass spectrometry can be used, which is especially handy when measurements are performed at remote locations (Barreira et al., 2015; Contreras et al., 2008).

In this study, the potential of NTME and portable gas chromatography-mass spectrometry for fast *in-situ* measurement of biogenic and anthropogenic VOCs was evaluated. The results obtained were compared with meteorological data and with particle number concentration at the measurement site.

## 2. Experimental methods

### 2.1 Chemicals and materials

CALION™ PV Mix standards, containing 13 compounds adsorbed into a granular solid matrix, were used for performance validation of portable GC-MS. Standard of dimethylamine (DMA) was prepared by diluting dimethylamine hydrochloride (99%) from Sigma-Aldrich (St. Louis, USA) with ultrapure water (DirectQ-UV, Millipore, USA). A volume of 5 mL was then added to a headspace vial of 20 mL and sealed with UltraClean 18 mm screwcap with septa (Agilent, Waldbronn, Germany). Sodium hydroxide (0.1 M) from FF-Chemicals Ab (Yli Ii, Finland) was added into the headspace vial with syringe to release DMA into the headspace. (+)-3-Carene ($\geq$ 98.5 %) and α-Pinene (98 %) from Sigma-Aldrich (St. Louis, USA), hexanal, heptanal, octanal, nonanal, decanal and benzaldehyde from Accustandard (New Heaven, CT, USA, purity not given), ethylbenzene (99 %) and p-xylene ($\geq$ 99 %) from Aldrich (Milwaukee, WI), and m-xylene ($\geq$ 99.3 %) from Merck (Darmstadt, Germany) were used as standards. Pinonaldehyde was synthesized by oxidative bond cleavage of pinanediol (Sigma Aldrich, St. Louis, USA, 99 %) according to the method described by Glasius et al. (1997). SFE-grade $CO_2$ (AGA, Espoo, Finland) was used for the determination of gas hold-up volume (time) ($t_M$)).

### 2.2 Measurement site

VOC sampling was made at the SMEAR II station (Station For Measuring Ecosystem-Atmosphere Relations, 61°50.845′ N, 24°17.686′ E, 179 m above sea level) at Hyytiälä, in southern Finland (Hari and Kulmala, 2005). Tampere, a city with around 200 000 inhabitants, is located 60 km southwest from the SMEAR II station. Flora in Hyytiälä is dominated by Scots pine and Norway spruces of approximately 50 years old and about 18 m canopy height. The sampling place was located about 15 meters from a 127 m high mast for atmospheric and flux measurements mounted 2 meters above the average forest floor. For comparison reason, the sampling system was installed in a fixed place, which was located at 30 cm height from the ground vegetation and at about 3 meters from the closest trees.



## 2.3 Sampling and Analysis

Samples were collected in mid-summer and autumn 2014. The first part of the sampling campaign occurred from 12th of June to 10th of July to test the applicability of NTME and portable GC-MS for qualitative and semi-quantitative measurement of VOCs. The second part of the campaign was held from 3rd to 12th of November to verify the effect of the snow pack on the

concentration of VOCs in the air. Sampling was performed by using CUSTODION® needle trap microextraction syringes (Torion Technologies Inc., Utah, USA). These syringes consist of 19-gauge stainless steel needles with a small side-hole for dynamic sampling and packed with Tenax TA (1 mg, 60-80 mesh), Carboxen 1016 (1.6 mg , 60-80 mesh) and Carboxen 1003 (1.5 mg, 60-80 mesh). Needle trap syringe was installed in a commercial air sampling interface made for needle trap devices (Torion Technologies Inc.). A BUCK Elite$^{TM}$ air sampling pump was used. Besides, a 0.45 μm nylon filter (Nalgene,

Rochester, NY) was implemented in the sampling system in order to prevent the blocking of the needle by aerosol particles and other impurities, without compromising the air flow. The air was filtrated through the filter directly into the needle trap device. Analytes were then dynamically extracted onto the packing material of the needle trap microextraction syringe. The sampling flow rate and time were 25 mL/min and 100 minutes, respectively. The studied compounds are found in trace levels in ambient air, which requires high sampling volumes to be concentrated. Thus, the breakthrough volume of the needle trap is

of concern. Determination of breakthrough volume is challenging as it depends on several conditions such as the type and amount of sorbent materials, the composition and concentration of the sample, temperature and relative humidity (Sanchez, 2014). For some sorbent materials, breakthrough values are given by manufacturers (e.g. http://www.sisweb.com/index/referenc/resins.htm). According to these reference values a sampling volume of 2.5 L is lower than the breakthrough volumes. This was also verified experimentally. One μL of α-pinene was evaporated by heating in a

headspace vial (20 mL), from which 0.5 mL of gas phase was transferred by gas-tight syringe to a Teflon bag (Tedlar®, 10 L, Sigma Aldrich) filled with nitrogen. Samples of 2, 2.5 and 3 L were collected sequentially by NTME and analysed by portable GC-MS.   A CUSTODION® solid phase microextraction syringe (Torion Technologies Inc., Utah, USA) containing a fiber coated with Divinylbenzene/Polydimethylsiloxane (DVB/PDMS, 65μm film thickness) was used to obtain retention times and mass spectra for the standards of the analytes tentatively identified with mass spectral library (NIST 2014) search. VOCs were

analysed on a portable GC-MS (TRIDION™-9 Torion Technologies Inc., Utah, USA) consisting of a low thermal mass injector, a low thermal mass capillary gas chromatograph (MXT-5 column, 5 m × 0.1 mm, 0.4 μm film thickness) and a miniature toroidal ion trap mass analyser. A monotherm heatable magnetic stirrer (Variomag Electronicrührer, Labortechnik, Munich, Germany) was used for the heating and stirring of the standards. Air samples were analysed by a similar method to the one described in Barreira et al. (2015). The NTME syringe was placed into the injection port of the portable GC-MS and

the needle trap was exposed during 10 seconds for thermal desorption. Injector and transfer line temperatures were set to 270°C and ion trap temperature to 150°C. A 10:1 split ratio was applied for 2 seconds after injection, being increased to 50:1 from 10 seconds to 30 seconds. The temperature program started from 50°C for 10 seconds, and  was increased to 270°C at 2°C/s. Analytes were ionized by electron ionization (70 eV) and delivered into the mass analyzer, which was set to scan a mass range





from 43-500 amu. The total run time was 180 seconds. The carrier gas was helium of 99.996 % purity (AGA, Espoo, Finland). A blank run was performed between each sample collection. For semi-quantitation, extracted ion chromatograms with base ions were used (m/z 93 for $\Delta^3$-carene and α-pinene; m/z 91 for ethylbenzene, m-xylene and p-xylene, m/z 83 for pinonaldehyde; m/z 105 for benzaldehyde; m/z 56 for hexanal; m/z 55 for heptanal, m/z 69 for octanal; m/z 57 for nonanal and 81 m/z decanal).

## 5  3. Results

Portable gas chromatography-mass spectrometry employed in this study has been developed for the analysis of VOCs in our previous research (Barreira et al., 2015). The system was further developed in this work by including needle trap microextraction to allow higher sampling capacity of VOCs. In contrast to our previous study, where solid-phase microextraction was used, NTME is more robust and enables exhaustive sampling. Factors affecting sensitivity and

chromatographic separation including peak shape were considered. Quantitation was challenging, mainly due to the difficulties in the preparation of gaseous calibration standards for VOCs at accurate concentrations. For this reason a semi-quantitative approach was used.

Due to small differences in the sampling volume (Tables S1 and S2) and because breakthrough was not observed for the sampling volume, the peak areas were divided by the volume of air collected to normalize the data. Temperature (measured at

4.2 m height) and particle number concentration (available at http://avaa.tdata.fi/web/smart and provided by Junninen et al., 2009) were used for comparison with obtained results. Temperature can influence VOC emissions and thus, must be taken in consideration for the interpretation of results. Between days temperature varied significantly (from 5 °C to 27 °C). However, during the days, it was almost constant (daily standard deviation less than 4 °C) and its effect can be neglected.

New particle formation events were classified according to Dal Maso et al. (2005), and the classification was further used to

evaluate the participation of studied compounds in particle formation and growth events during the first part of the campaign. This classification divided nucleation days into event, non-event and undefined. Event days are those when there is a distinctly new mode of particles starting in the nucleation mode, which prevails and grows over hours. On undefined days, particles are occasionally formed in the nucleation mode range or there is a later phase of a mode growing in the Aitken-mode size range. Non-event days are those where the referred conditions were not observed. A total of 7 event days, 12 undefined event days

and 5 non-event days were observed.

### 3.1 Qualitative Analysis

Numerous compounds were tentatively identified from the samples using NIST 2014 spectral library search. Target compounds included monoterpenes, such as α-pinene and $\Delta^3$-carene, aldehydes, amines and other compounds with recognized relevance for atmospheric physics and chemistry. For the selected compounds, identification was confirmed with authentic standards.

Standard samples were prepared by total evaporation of 1 μL of each standard in a headspace vial, transferred by gas-tight syringe to another headspace vial for further dilution, collected by SPME (PDMS/DVB, 65 μm of film thickness) and analysed





by portable GC-MS. In contrast, DMA samples were made according to the procedure explained in section 2.1. The retention times and most abundant fragments obtained by portable GC-MS are presented in Table 1. A slight difference in fragmentation was found between the conventional and portable GC-MS, the difference being more expressive for the compounds that are more extensively fragmented. The knowledge of the most abundant masses found by portable GC-MS can then be defined as

a critical factor during semi-quantitative/quantitative analysis of compounds with high fragmentation. Thus, the use of standards during qualitative analysis allowed the proper identification of compounds with different properties by NTME and portable GC-MS, which in turn provided a fast in-situ method for the identification of atmospherically relevant gaseous compounds.

### 3.2 VOCs in air samples

Monoterpenes and their oxidation products are an important group of compounds that can participate in aerosol formation and growth (e.g. Spracklen et al., 2006). In this study, α-pinene, $\Delta^3$-carene and pinonaldehyde were measured by NTME-GC-MS (more detailed information can be found in Table S1). First two compounds have been reported as the main emitted monoterpenes at the boreal forest of Hyytiälä (e.g. Yassaa et al., 2012). Fig. 1 shows the comparison between measured amounts of α-pinene, $\Delta^3$-carene and pinonaldehyde. Temperature was included for the comparison, since temperature

dependent emissions have been reported in many previous studies (e.g. Tarvainen et al., 2005). This phenomenon was also observed in the obtained results, since a significant increase in temperature was followed by an increase in the amount of measured monoterpenes in the air. A similar trend on the amounts of α-pinene and $\Delta^3$-carene was observed, which can be explained by the fact that these compounds are both emitted by *Pinus sylvestris,* the dominant tree species at Hyytiälä boreal forest (Rinne et al., 2000). The results are also consistent with a previous study performed by the same type of NTME syringes

and portable GC-MS, used for the *in-situ* headspace collection and analysis of volatiles from varying yellow starthistle flower heads, sealed in a modified scintillation vial collection apparatus (Beck et al., 2015).

The most significant oxidation product in the air samples was pinonaldehyde. This observation is related to its vapor pressure that is higher than that of other oxidation products such as pinonic acid, which are more likely to be in the particulate phase. A moderate correlation coefficient (0.52) was found between α-pinene and pinonaldehyde. This result was expected, since an

25 increase in the amount of precursor compounds results in higher output of oxidation products, when optimum conditions that favor the reaction are met. However, pinonaldehyde was detected only in a few samples. For that reason, further development of the method is still required. The importance of α-pinene, $\Delta^3$-carene and pinonaldehyde for atmospheric particle formation and growth has been emphasized also in previous studies, revealing the need for continuous measurement (e.g. Kavouras et al., 1998).

Aldehydes are another class of compounds with a remarkable importance to atmospheric chemistry, due to their photolytic reactions in the atmosphere and contribution to new particle formation (e.g. Jang and Kamens, 2001). As shown in Fig. 2, several aldehydes were found in the samples, in particular hexanal, heptanal, octanal, nonanal, decanal and benzaldehyde (more detailed information can be found on Table S2). These compounds were also found in another study performed in the





boreal forest of Hyytiälä. The authors used $C_{18}$-cartridges coated with DNPH (2,4-dinitrophenyl hydrazine) for collection and LC-MS analysis (Hellén et al., 2004) to evaluate the effect of temperature, and found to have some effect on the amount of aldehydes measured in the air. However, the temperature effect in our study cannot explain per se the variance. Diversity of factors, instead, which can also influence the emission of VOCs, are most likely the reason. The correlation between aldehydes,

including pinonaldehyde, was evaluated. A good correlation was found between studied aldehydes (Table S3). The results suggest that these aldehyde emissions originate from the same source. Pinonaldehyde, as an exception, is indicative of a distinct and well known formation source consisting of the α-pinene atmospheric oxidation. Because other aldehydes are not formed as a result of oxidative processes in the air, their common source is most probably associated with the vegetation itself. Aldehyde emission from vegetation has been found in other studies (e.g. Wildt et al., 2003) also, using similar NTME syringes

and portable GC-MS (Beck et al., 2015).

Ethylbenzene and para- and meta-xylenes were also identified in the collected samples (Fig. 3, for more detailed information Table S1). These anthropogenic SOA precursors can be important to boreal forest environment and, in addition, they are considered harmful for human health and vegetation, and they can interfere with natural atmospheric processes (e.g. Paralovo et al., 2016). Air mass origins were determined using the HYSPLIT transport and dispersion model from the NOAA Air

Resources Laboratory (Stein et al., 2015). These compounds were mainly transported from cities such as Tampere with relevant anthropogenic activity (e.g. Fig. S1 (A)). On the other hand, no pollution events were observed during the days when the wind was coming from geographic coordinates that are relatively pristine, compared to the referred cities, such as the regions located in the north of Finland (e.g. Fig. S1 (B)). More investigation about the effect of these compounds on the climate is still needed, and the method used in this study can constitute an easy alternative and/or a complementary system to conventional methods.

Dimethylamine (DMA) was tentatively identified in this study. Characteristic ions and retention time similar to that obtained for standard were found with the method used. However, without almost any retention in the column, and due to possible chromatographic overlapping with ethylamine (EA) and/or $CO_2$, further development of the method for their analysis is still needed.

### 3.3 VOCs and particle number concentration

The influence of BVOCs on new particle formation was evaluated in this study. The type of nucleation event and particle number concentration were considered in the evaluation. As demonstrated in Fig. 4, particle number concentration was significantly higher during event days when compared to undefined and non-event days. However, the amounts of monoterpenes and pinonaldehyde (amounts were multiplied by a factor of 10 for a better visualization) were lower during days when particle number concentration increased (Fig. 4 (A)). This observation clearly indicates the oxidation of monoterpenes

and subsequent partition of their oxidation products in particulate phase. Also, aldehyde concentrations were lower when particle number concentration was higher (Fig. 4 (B)). This evidence was expected since aldehydes can also contribute to new particle formation and growth (e.g. Jang and Kamens, 2001). Altogether, our findings are in agreement with observations



found in other studies that demonstrated the participation of VOCs in atmospheric particle formation and growth (e.g. Laaksonen et al., 2008).

## 3.4 VOCs during a snow melt event

The amounts of biogenic volatile compounds were studied by NTME-GC-MS before a soil snow coverage and when snow melted in November. The purpose of the study was to understand how this phenomena affect the concentrations of the atmospheric BVOCs. Also, it can influence the measured atmospheric levels of the studied compounds at soil level.

Fig. 5 shows the atmospheric amounts of monoterpene and aldehydes during the snow episode. Monoterpenes, and in particular α-pinene and $\Delta^3$-carene, were found in small amounts in the beginning of the campaign before the snow event, as expected during this time of the year at the boreal forest site. Temperature was higher during the first part of the campaign, indicating more favorable conditions for higher monoterpene amounts when compared to colder days, as previously described in other studies (e.g. Song et al., 2014; Tingey et al., 1980). During the snow event, no measurements were performed due to technical limitations (needle trap could freeze and block because of negative temperatures). On the 9th of November, when the temperature raised above 0°C, monoterpenes were still not found, but they increased rapidly on the 10th of November during the snow melt overcoming the amounts found before the snow at higher temperatures. Altogether, these observations suggest the accumulation of these compounds under or into the snow pack. The same phenomena have been verified in another study (Aaltonen et al., 2012). A similar comparison was performed for aldehydes identified and measured in the same sampling period. For the comparison, all identified aldehydes during the autumn campaign were considered. Once more, aldehydes were found in lower amounts during the beginning of the campaign when the temperature was higher. After the snow melt, aldehydes increased markedly reaching levels that overcome the amounts found in the warmest period of the November campaign. This evidence reveals the occurrence of an analogous phenomenon as the one observed for monoterpenes, suggesting that not only monoterpenes but also other BVOCs may undergo an accumulation in the snow pack during snow fall, being released when snow melts. The pool storage of these compounds can be particularly important in spring, when nucleation events are more frequent (e.g. Dal Maso et al., 2005). A snow accumulation of BVOCs during winter and their subsequent release during spring time could contribute significantly to new secondary organic aerosol formation by photo-oxidative processes, which reinforces the interest of studying further these phenomena.

## 4. Conclusion

VOCs were collected at the boreal forest (SMEARII, Hyytiälä, Finland) by NTME, and analysed by portable GC-MS. Neither additional sampling line nor sample pre-treatment was needed, which reduced analysis time, sample contamination and potential losses during analytical process. The most abundant monoterpenes, pinonaldehyde and aldehydes were successfully measured. The amounts of monoterpenes and aldehydes seem to be lower during days when particle formation was more prone to happen. Ethylbenzene, para- and meta-xylene were also found, when wind direction was from the cities with significant



anthropogenic activity. Increased monoterpene and aldehyde amounts were verified after a snow melt event, revealing their accumulation under or into the snow pack. Results demonstrated the potential of the method for the rapid *in-situ* sampling and analysis of volatile organic compounds in the atmosphere, which can be favorable for the chemical characterization of atmosphere at remote places. Semi quantitative approach proved to be promising, but quantification with the method

5 developed, including determination of breakthrough volume and addition of internal standard, requires additional studies.

**Acknowledgements**

Financial support was provided by the Academy of Finland Centre of Excellence program (project no 272041). Torion technologies Inc. (Utah, USA) is thanked for the cooperation. The authors gratefully acknowledge the NOAA Air Resources Laboratory (ARL) for the provision of the HYSPLIT transport and dispersion model and/or READY website

10 (http://www.ready.noaa.gov) used in this publication. The staff of Laboratory of Analytical Chemistry and Smear II station are thanked for the cooperation. Jenni Kontkanen is particularly thanked for providing the new particle formation events classification.





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



**Table 1: Retention times and most abundant fragments (in order of descending intensity) obtained by portable GC-MS for the identified compounds.**

| Compound | R.T. [min] | Fragmentation pattern with portable GC-MS | Fragmentation pattern obtained from NIST library |
|---|---|---|---|
| *Monoterpenes* | | | |
| α-pinene | 0.82 | 93 (100), 92, 91, 53, 79 | 93 (100), 91, 92, 39, 77 |
| Δ³-carene | 0.93 | 93 (100), 91, 79, 92, 80 | 93 (100), 91, 92, 39, 77 |
| *Aldehydes* | | | |
| benzaldehyde | 0.87 | 105 (100), 53, 107, 106, 77 | 77 (100), 106, 105, 51, 50 |
| hexanal | 0.60 | 56 (100), 57, 67, 83,72 | 44 (100), 56, 41, 43, 57 |
| heptanal | 0.76 | 55 (100), 71, 97, 70, 44 | 70 (100), 41, 44, 43, 55 |
| octanal | 0.91 | 69 (100), 111, 57, 67, 56 | 43 (100), 44, 41, 56, 84 |
| nonanal | 1.05 | 57 (100), 143, 67, 69, 81 | 57 (100), 41, 43, 56, 44 |
| decanal | 1.19 | 81 (100), 67, 83, 69, 57 | 43 (100), 41, 57, 55, 44 |
| pinonaldehyde | 1.30 | 151 (100), 83, 107, 97, 109 | 83 (100), 69, 43, 98, 55 |
| *BTEX* | | | |
| ethylbenzene | 0.70 | 91 (100), 106, 92, 78, 65 | 91 (100), 106, 51, 65, 77 |
| p-xylene | 0.72 | 91 (100), 106, 105, 79, 119 | 91 (100), 106, 105, 77, 51 |
| m-xylene | 0.72 | 91 (100), 106, 105, 107,65 | 91 (100), 106, 105, 77, 51 |
| *Amines* | | | |
| dimethylamine | 0.17* | 44 (100), 46, 45, 43, 42 | 44 (100), 45, 28, 42, 43 |

*\* no retention in the column ($t_M$=0.12min).*

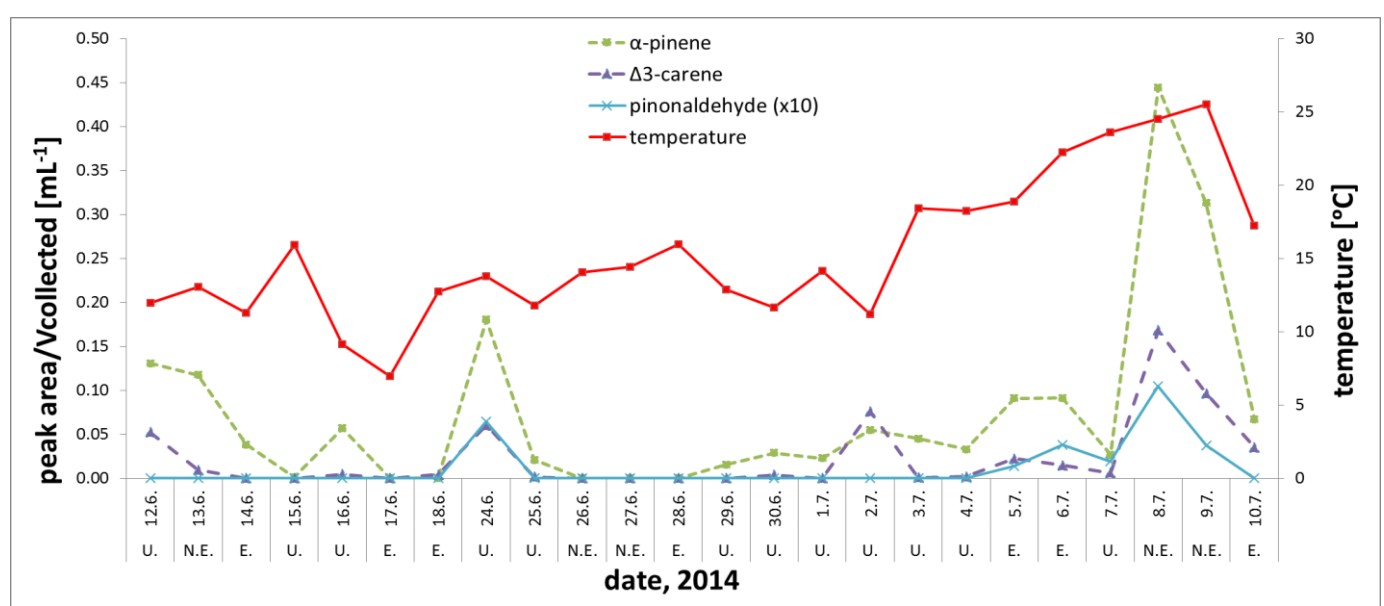

**Figure 1: Comparison between the daily average amounts (peak areas/Vcollected) of α-pinene, Δ³-carene and pinonaldehyde (Note the different scale for pinonaldehyde (10x)) collected by NTME and analysed by portable GC-MS in mid-summer 2014. (E. corresponds to a day when a nucleation event was observed, U. to an undefined event and N.E. to a day without nucleation events).**





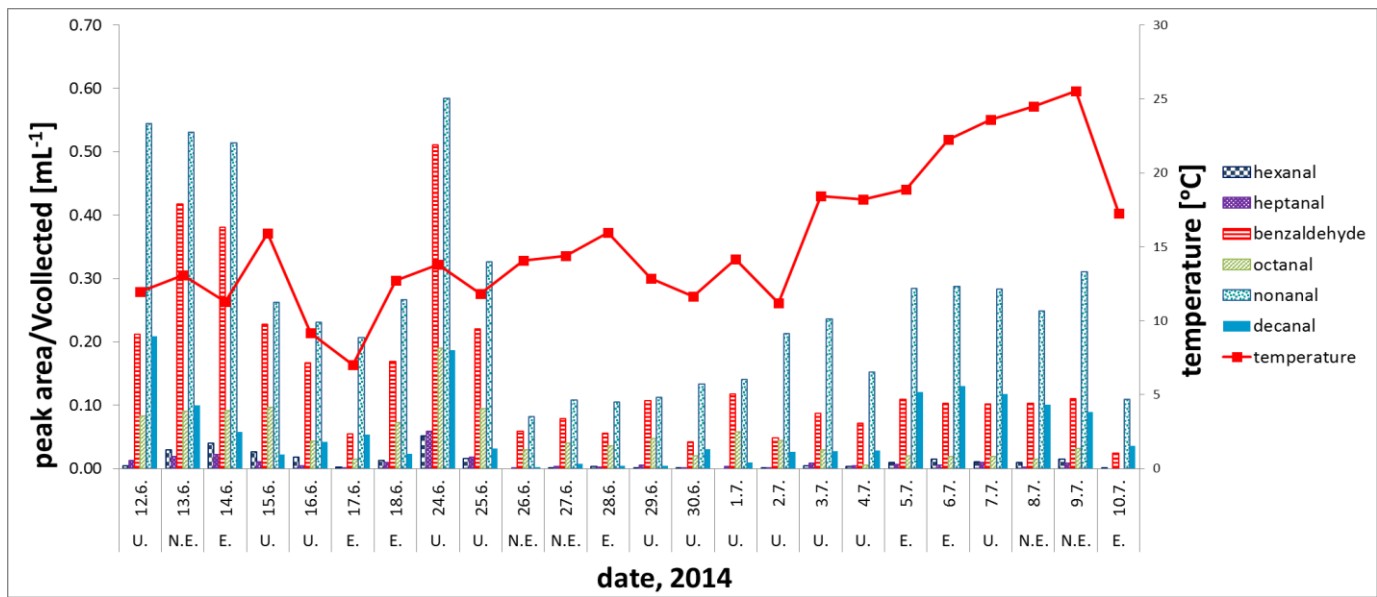

**Figure 2: Daily average amounts (peak areas/Vcollected) of aldehydes collected by NTME and analysed by portable GC-MS in mid-summer 2014 (E. corresponds to a day when a nucleation event was observed, U. to an undefined event and N.E. to a day without nucleation events).**

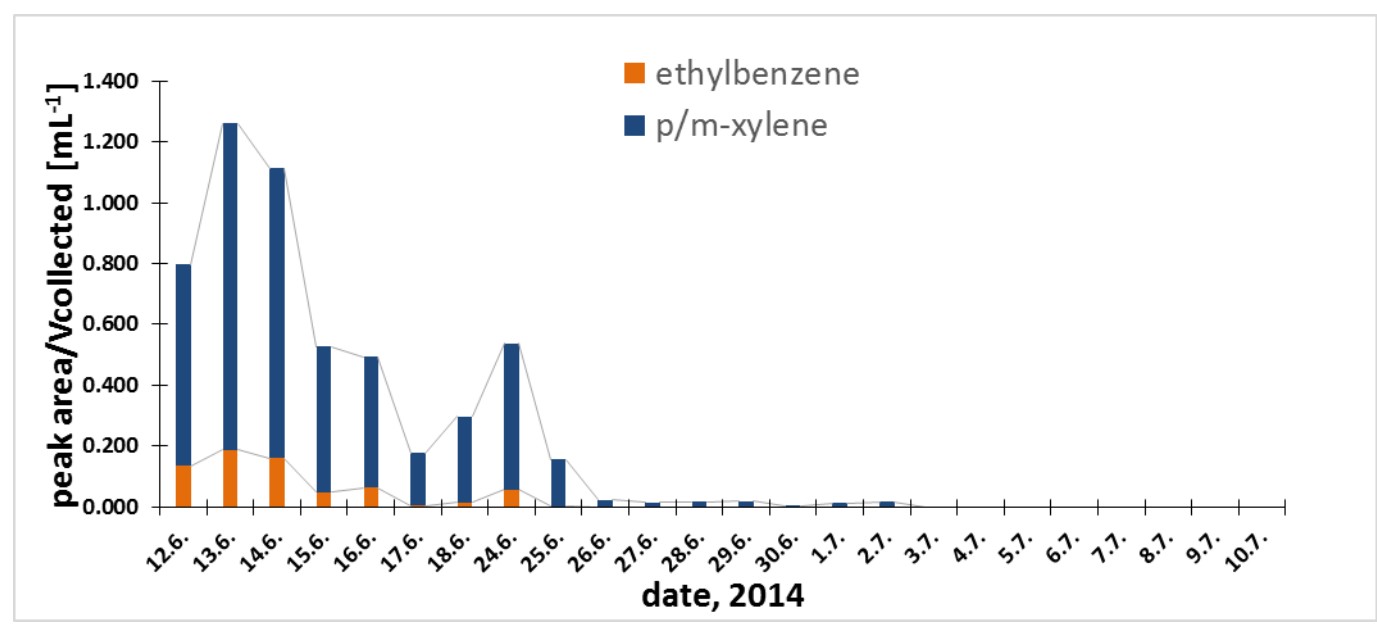

**Figure 3: Occurrence of pollution events during the summer campaign, 2014. Ethylbenzene and p/m-xylene were identified and monitored by NTME and portable GC-MS during the campaign period.**



**Figure 4: Comparison between the particle number concentration (#/cm³) and the daily average amounts (peak areas/Vcollected) of α-pinene, pinonaldehyde (A), and the sum of aldehydes (benzaldehyde, hexanal, heptanal, octanal, nonanal, and decanal) (B) collected by NTME and analysed by portable GC-MS in mid-summer 2014. (E. corresponds to a day when a nucleation event was observed, U. to an undefined event and N.E. to a day without nucleation events).**



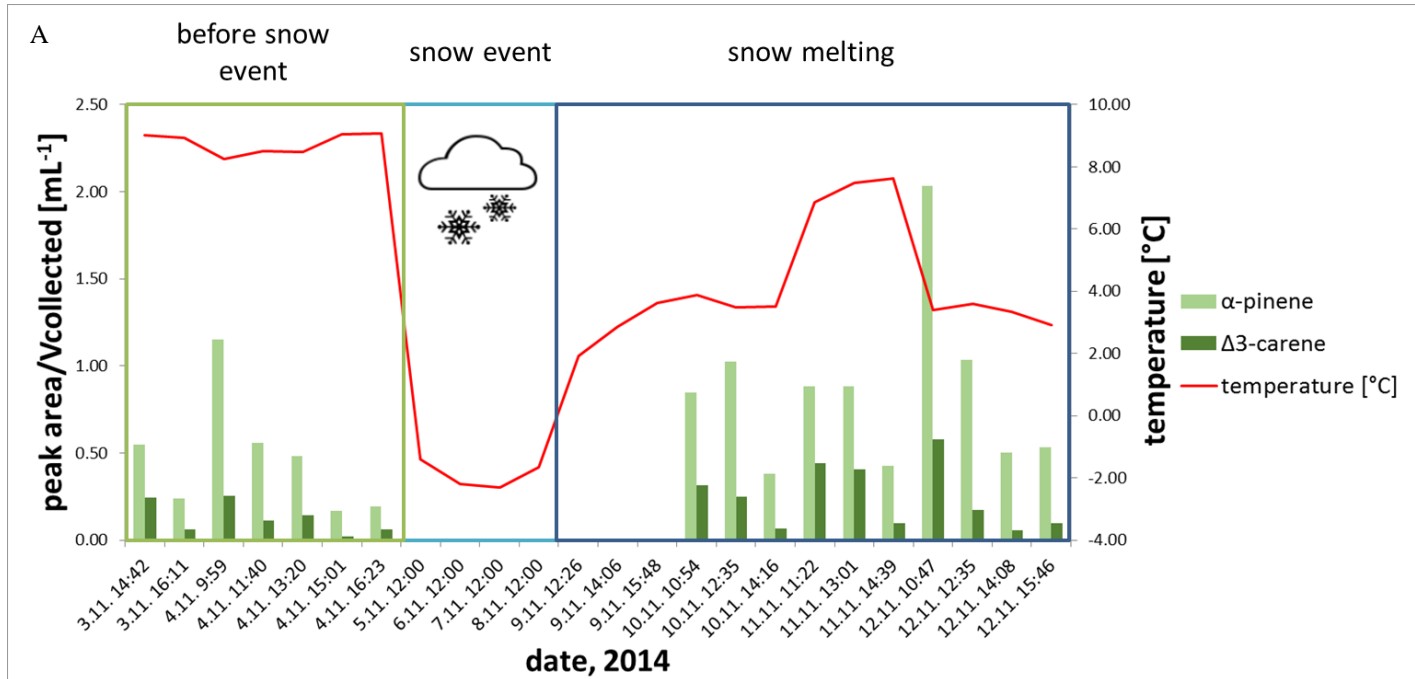

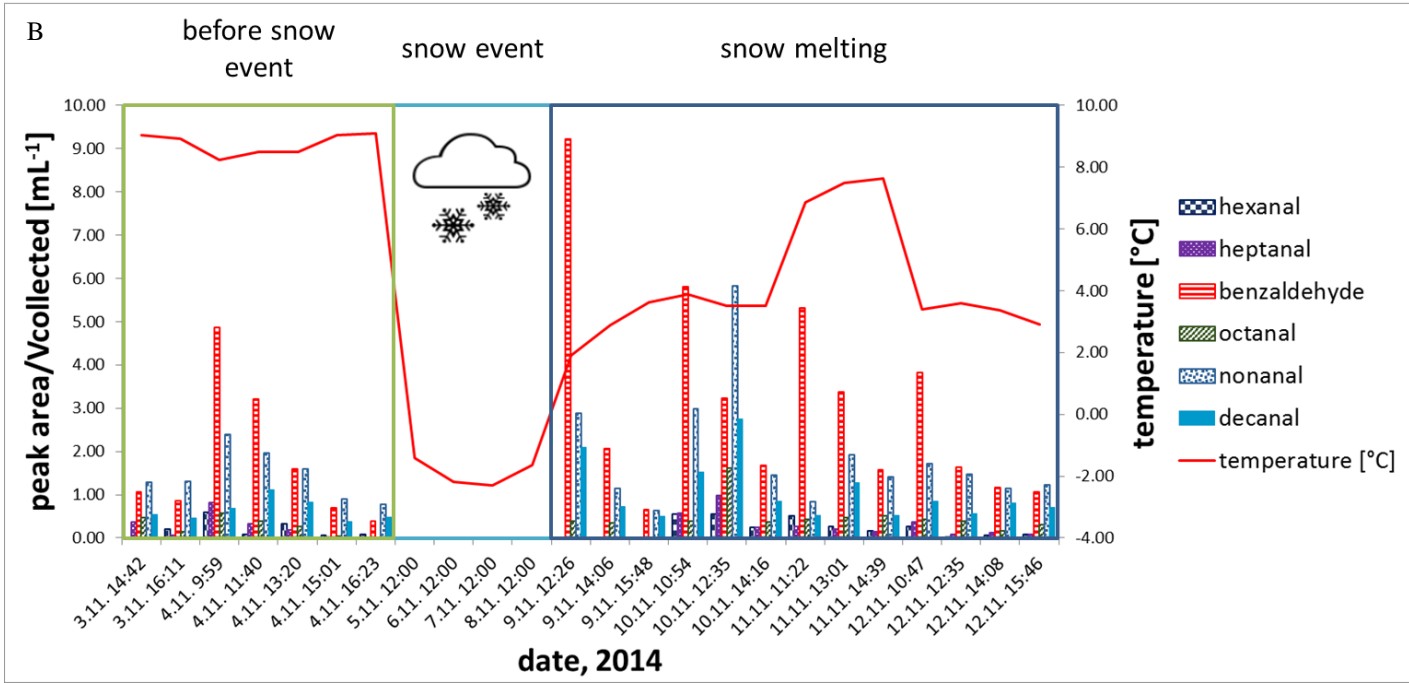

**Figure 5: Observation of an increase in monoterpene (A) and aldehyde (B) atmospheric amounts during a snow melt event in November 2014 (Daily average temperatures were used on days when sampling was not performed).**

