# Peer review of "Potential of needle trap microextraction – portable gas chromatography – mass spectrometry for measurement of atmospheric volatile compounds"

_Atmospheric Measurement Techniques, 2016_

## Referee Comment (RC1) · W Filipiak (Referee) · 12 Apr 2016

Referee comments

1. General Comments

The manuscript of Barreira et al., entitled "Potential of needle trap microextraction– portable gas chromatography–mass spectrometry for measurement of atmospheric volatile compounds" addresses the important question of application of modern analytical techniques in the atmospheric measurements. Combination of Needle Trap Microextraction (NTME) technique with portable GC-MS provides novel approach for

fast and in situ analysis of target analytes in environmental samples. The results are good and support the assumption of application of portable NTME-GC-MS system for Biogenic VOC analysis in atmospheric air. Authors demonstrated that using the here proposed method it is possible to link the concentrations of diverse organic molecules (mainly aldehydes) with new particle formation and provide an evidence that Biogenic VOCs can be accumulated in and later released from the snow, potentially contributing to the formation of secondary organic aerosol.

2. Specific Comments

Authors should consider addition of more references to the NTME technique. Apart from environmental analysis, there are several interesting articles about NTME dealing with optimization of adsorption and desorption parameters affecting the efficient pre-concentration of volatile organic compound. In this regard, authors may consider the following additions/clarifications:

- Introduction, p. 2, line 34: after the sentence ". . . of the sorbent packed in the needle (Eom et al., 2008)." add the information that - additionally to the mentioned volume of adsorbent - also the type of adsorbent material (microporosus/mesoporous structure, mechanical/thermal stability) as well as sampling parameters (temperature and sample flow rate during adsorption) affect the reproducibility and efficiency of adsorptive preconcentration on needle trap devices as demonstrated by Filipiak et al. [doi:10.1088/1752-7155/6/2/027107 J. Breath Res. 6 (2012) 027107 ]. Furthermore, the robustness, easiness and rapidity of the analysis with NTME were shown to be superior for the BTEX determination in gaseous and even in aqueous samples [Jurdakova et al., doi:10.1016/j.chroma.2008.04.065 J. Chrom. A, 1194, 2008, 161–164].

- Section 2.3, p. 3, line 30: it would be helpful to explain to a reader what exactly is "CUSTODION® needle trap microextraction syringe", as there is no information about this device on a manufacturer's website (http://torion.com/products/custodion.html). It is particularly interesting exactly how was a thermal desorption performed, as authors
mention "syringe" in terms of needle trap device (NTD), whereas desorption from needle traps is typically done by simple insertion of NTD (plugged on other end) into GC injector (operating at preset temperature)... Did authors use additional sample-flow through the needle trap during injection to GC?

- Section 2.3, p. 4, line 8: authors used 25 ml/min sample flow rate over 100 minutes for adsorption on Needle Trap Device (NTD), what results in 2,5 L of sample drag through needle trap. Are the authors sure that there is no saturation of adsorbents in needle trap? This is very large volume of sample, typically used for adsorption on conventional "sorption tubes" (sampling tubes) filled with incomparably larger amounts of the same adsorbents...

- Section 2.3, p. 4, line 23: in extracted ion chromatograms it has no sense to use m/z=83 for propionaldehyde. This compound has a molecular weight of 58 and simply cannot generate a signal at m/z=83.

- Section 3, p. 4, line 30: if authors state that "factors affecting sensitivity and chromatographic separation including peak shape were considered", they should also provide information which exactly factors were taken into consideration, what is the effect, why certain parameters (what values) were selected etc. In a present form such statement does not bring much information...

- Section 3, p. 4, line 32: authors normalized the peak areas to the adsorbed gas volume. This could be done only in case of linear relationship of the acquired peak area and adsorbed sample volume (i.e. no saturation of NTD). Could authors present a proof that collecting 2,5L of sample on needle trap still guarantees this linearity?

3. Technical Correction

Authors are asked to clarify the description of sampling procedure in the following way: a) If there was no syringe (containing a sample) connected to a needle trap, authors should not use the term "needle trap syringe" (e.g. p.4, line 3) but "needle trap device"

or an abbreviation "NTD" thorough the manuscript. b) If there was a syringe (with sample) connected to a needle trap, it should be stated clearer (but this would mean 2,5L syringe, according to author's description of sampling). Perhaps, authors could add a sketch depicting the system used.

---

## Author Comment (AC1) · 3 Jun 2016

Answers to reviewer's comments

(1) Authors should consider addition of more references to the NTME technique. Apart from environmental analysis, there are several interesting articles about NTME dealing with optimization of adsorption and desorption parameters affecting the efficient preconcentration of volatile organic compound. In this regard, authors may consider the following additions/clarifications: - Introduction, p. 2, line 34: after the sentence ": : : of the sorbent packed in the needle (Eom et al., 2008)." add the information

that - additionally to the mentioned volume of adsorbent - also the type of adsorbent material (microporosus/mesoporous structure, mechanical/thermal stability) as well as sampling parameters (temperature and sample flow rate during adsorption) affect the reproducibility and efficiency of adsorptive preconcentration on needle trap devices as demonstrated by Filipiak et al. [doi:10.1088/1752-7155/6/2/027107 J. Breath Res. 6 (2012) 027107 ]. Furthermore, the robustness, easiness and rapidity of the analysis with NTME were shown to be superior for the BTEX determination in gaseous and even in aqueous samples [Jurdakova et al., doi:10.1016/j.chroma.2008.04.065 J. Chrom. A, 1194, 2008, 161–164].

(2) The references were added to the manuscript.

(3) Page 3, line 3: It was demonstrated that NTME was easy, fast and robust in a study on determination of trace amounts of BTEX in water by GC-FID (Jurdáková et al., 2008). The type of adsorbent material (microporous/mesoporous, mechanical/thermal stability) and sampling parameters (such as temperature and sample flow during adsorption) affect the efficiency and reproducibility of adsorption on needle trap devices (Filipiak et al., 2012).

(1) Section 2.3, p. 3, line 30: it would be helpful to explain to a reader what exactly is "CUSTODION$^{®}$ needle trap microextraction syringe", as there is no information about this device on a manufacturer's website (http://torion.com/products/custodion.html). It is particularly interesting exactly how was a thermal desorption performed, as authors mention "syringe" in terms of needle trap device (NTD), whereas desorption from needle traps is typically done by simple insertion of NTD (plugged on other end) into GC injector (operating at preset temperature): : : Did authors use additional sample-flow through the needle trap during injection to GC?

(2) Manufacturer's reference on NTD can be found at http://torion.com/fileadmin/media/documents/applications/TLPN_1245_Torion_Application_Brief_VOC_s_in_Air_02_01_1: Thermal desorption was performed by inserting the needle trap device into the in-

jection port of the GC-MS (at 270 °C). External flow of carrier gas was not needed. A more detailed description about the thermal desorption process was added to the manuscript.

(3) Page 5, line 7: The injector assembly allows for a proper flow of carrier gas through the side-hole placed above the adsorbent material in the needle trap, which directs the desorbed compounds into the GC column during thermal desorption.

(1) Section 2.3, p. 4, line 8: authors used 25 ml/min sample flow rate over 100 minutes for adsorption on Needle Trap Device (NTD), what results in 2,5 L of sample drag through needle trap. Are the authors sure that there is no saturation of adsorbents in needle trap? This is very large volume of sample, typically used for adsorption on conventional "sorption tubes" (sampling tubes) filled with incomparably larger amounts of the same adsorbents. . .

(2) An additional study on the needle trap breakthrough volume was added to the manuscript. No saturation was observed for a concentration of 386 ppbV of $\alpha$-pinene, which is significantly higher than ones usually observed for monoterpenes, aromatics and carbonyl compounds in Hyytiälä ambient air (eg. Hellén et al., 2004). (3) Page 4, line 24: To ensure sufficient capacity of the sampler and that the breakthrough volume is not reached during sampling of 2.5 L of air, following study was performed. One $\mu$L of $\alpha$-pinene was evaporated by heating in a headspace vial (20 mL), from which 0.5 mL of gas phase was transferred by gas-tight syringe to a Teflon bag (Tedlar$^{®}$, 10 L, Sigma Aldrich) filled with nitrogen. The final concentration was 386 ppbV, which is significantly higher than the concentrations usually measured in the ambient air (e.g. Hellén et al., 2004). A sampling kit (Pas Technology, Germany) was used for collection of different volumes of $\alpha$-pinene from the Tedlar bag into the needle trap device, and samples were analyzed by GC-MS. As can be seen in Fig. S2, a good linearity was observed for all volumes sampled, which strongly suggest that saturation or breakthrough volume was not achieved during the field measurements. (1) Section 2.3, p. 4, line 23: in extracted ion chromatograms it has no sense to use m/z=83 for propionaldehyde. This compound

has a molecular weight of 58 and simply cannot generate a signal at m/z=83.

(2) The compound is pinonaldehyde (C10H16O2, MW = 168.23 g/mol), an oxidation product of $\alpha$-pinene.

(3) No changes has been made.

(1) Section 3, p. 4, line 30: if authors state that "factors affecting sensitivity and chromatographic separation including peak shape were considered", they should also provide information which exactly factors were taken into consideration, what is the effect, why certain parameters (what values) were selected etc. In a present form such statement does not bring much information...

(2) Collection time, split ratio and duration were optimized. The choice of collection time was already justified by the trace levels of studied compounds found in the ambient air. The changes on the previous developed method for an increase in sensitivity, without compromising the chromatographic separation, are now described in the manuscript.

(3) Page 5, line 31: Portable gas chromatography-mass spectrometry method employed in this study has been developed for the analysis of VOCs in our previous research (Barreira et al., 2015). The system was further developed in this work by including needle trap microextraction to allow higher sampling capacity of VOCs. In contrast to our previous study, where solid-phase microextraction was used, NTME is more robust and enables exhaustive sampling. Minor modifications on the previous method, such as shorter split time, were performed in order to improve sensitivity without compromising the peak shapes.

(1) Section 3, p. 4, line 32: authors normalized the peak areas to the adsorbed gas volume. This could be done only in case of linear relationship of the acquired peak area and adsorbed sample volume (i.e. no saturation of NTD). Could authors present a proof that collecting 2,5L of sample on needle trap still guarantees this linearity?

(2) The linear relationship between acquired peak area and adsorbed sample volume

was demonstrated in the study on the breakthrough volume, which was added to the manuscript.

(3) Page 4, line 24: To ensure sufficient capacity of the sampler and that the breakthrough volume is not reached during sampling of 2.5 L of air, following study was performed. One $\mu$L of $\alpha$-pinene was evaporated by heating in a headspace vial (20 mL), from which 0.5 mL of gas phase was transferred by gas-tight syringe to a Teflon bag (Tedlar®, 10 L, Sigma Aldrich) filled with nitrogen. The final concentration was 386 ppbV, which is significantly higher than the concentrations usually measured in the ambient air (e.g. Hellén et al., 2004). A sampling kit (Pas Technology, Germany) was used for collection of different volumes of $\alpha$-pinene from the Tedlar bag into the needle trap device, and samples were analyzed by GC-MS. As can be seen in Fig. S2, a good linearity was observed for all volumes sampled, which strongly suggest that saturation or breakthrough volume was not achieved during the field measurements. (1) 3. Technical Correction Authors are asked to clarify the description of sampling procedure in the following way: If there was no syringe (containing a sample) connected to a needle trap, authors should not use the term "needle trap syringe" (e.g. p.4, line 3) but "needle trap device" or an abbreviation "NTD" thorough the manuscript. b) If there was a syringe (with sample) connected to a needle trap, it should be stated clearer (but this would mean 2,5L syringe, according to author's description of sampling). Perhaps, authors could add a sketch depicting the system used.

(2) The term "needle trap syringe" was corrected to needle trap device. A sketch depicting the system was added to the manuscript.

(3) Page 4, line 10: Needle trap device was installed in a commercial air sampling interface (Fig. S1) made for needle trap devices (Torion Technologies Inc.).
* * *
**Supplemental material for "Potential of needle trap microextraction – portable gas chromatography – mass spectrometry for measurement of atmospheric volatile compounds"**

L. M. F. Barreira[1], Y. Xue[1], G. Duporté[1], J. Parshintsev[1], K. Hartonen[1], M. Jussila[1], M. Kulmala[2], M.-L. Riekkola[1]

[1]Laboratory of Analytical Chemistry, Department of Chemistry, University of Helsinki, P.O. Box 55, FI-00014 Helsinki, Finland
[2]Division of Atmospheric Sciences, Department of Physics, University of Helsinki, P.O. Box 64, FI-00014 Helsinki, Finland

*Correspondence to*: M.-L. Riekkola (marja-liisa.riekkola@helsinki.fi)

[Figure]

**Figure S1: Schematic representation of collection system used for needle trap microextraction device.**

[Figure]

**Figure S2: Extraction volume vs peak area of α-pinene obtained by NTME and GC-MS.**

**Fig. 1.**

---

## Referee Comment (RC2) · Anonymous Referee #3 · 6 Jul 2016

Referee comments

1. General comments

This paper introduces the development and the application of an analytical technique based on a needle trap micro extraction coupled with a portable gas chromatography to the determination of organic gaseous compounds in the atmosphere. The advantages of needle trap micro extraction to the sampling of VOCs compared to a solid phase micro extraction has been described. Portable GC/MS instrumentation can achieve fast separation, identification and quantification of sample on site without the need

for transport to the laboratory. This minimizes the effects of volatiles lost and sample degradation during storage time. The results presented by the authors, demonstrated the successful applicability of this method for a semi quantitative approach. Indeed a quantification with the method developed in this manuscript requires additional studies. The manuscript is very interesting and the experiments seem to have been performed with rigor.

2. Specific comments

- I would add some references in the introduction, in particular page 3 – line 1 after ". . .since the sorbent particles are protected inside the needle" - Would it be possible to use a continuous analyzer of BTEX to compare the concentrations of this compounds founds by your method? - Section 2.2, page 3 – line 25: add some indications about the typical emission on the city of Tampere, allow to understand the situation around your sampling site. - Section 2.3, page 4 – line 13: I understand the need of a high sampling volume however are you certain not to saturate the adsorbents in needle trap. - Section 2.3, page 4 – line 22 : you use a DVB/PDMS for the solid phase micro extraction step, have you tested other fibers ? - Section 2.3, page 4 – line 32: the temperature program started from 50°C for 10 seconds and was increased to 270°C at 2°C/s. You write the total run time was 180 seconds, is there a problem in the total time? - Section 2.3, page 5 – line 4: after every analysis are you sure that all the compounds is totally desorbed? - Section 2.3,page 4: why didn't you use an internal standard for the quantification. - Section 3, page 5 – line 15: are the measurement of temperature and particle number concentration co- located with the sampling site? - Section 3.1, page 6 – line 3: have you tested the difference in fragmentation found between a conventional GC-MS and portable GC-MS or it's an observation from a bibliography? - Figure 1, 2,3 and 4: are these the average temperature during the day or during the period of sampling ? - Figure 3 : you quantified separately the ethylbenzene and the p/m- xylene, these two compounds that are well separated on your column ? (Tr = 0,70 min and Tr = 0,72 min). Which is the factor of resolution between these two compounds? Maybe you can

considered changing your column to improve the resolution.

---

## Author Comment (AC2) · 8 Jul 2016

(1) I would add some references in the introduction, in particular page 3 – line 1 after ". . .since the sorbent particles are protected inside the needle"

(2) A reference was added to the referred sentence.

(3) Page 2, line 34: VOCs can also be sampled by needle trap microextraction (NTME) that is considered to be more robust than other extraction techniques such as solid-phase microextraction, since the sorbent particles are protected inside the needle (Eom et al., 2008). Furthermore, because NTME is an exhaustive technique, the sensitivity

can be improved by increasing the sample volume; and capacity can be expanded by increasing the volume of the sorbent packed in the needle (Eom et al., 2012; Eom et al., 2008).

(1) Would it be possible to use a continuous analyzer of BTEX to compare the concentrations of this compounds founds by your method?

(2) A continuous analyzer of BTEX was not used during this study, but would be probably useful for comparison with results obtained for BTEX by portable GC-MS.

(3) No changes has been made.

(1) Section 2.2, page 3 – line 25: add some indications about the typical emission on the city of Tampere, allow to understand the situation around your sampling site.

(2) Major sources of air pollutants from urban areas in Finland include wood combustion and traffic.

(3) Page 3, line 29: Major sources of air pollutants from urban areas in Finland include wood combustion and traffic (e.g. Hellén et al., 2008; Taimisto et al., 2011).

(1) Section 2.3, page 4 – line 13: I understand the need of a high sampling volume however are you certain not to saturate the adsorbents in needle trap.

(2) In the corrected version of the manuscript after the first revision, information about breakthrough volume was added. Saturation was not observed for the concentrations typically founded during this study.

(3) No changes has been made, since they are already given in the previously modified manuscript version (apparently not visible for the referee #3).

(1) Section 2.3, page 4 – line 22 : you use a DVB/PDMS for the solid phase micro extraction step, have you tested other fibers ?

(2) We did not test other fibers because SPME was only applied for the verification of

studied compounds retention times and fragmentation in ion trap with authentic standards, thus any fiber with affinity for the studied compounds is suitable for this purpose.

(3) No changes has been made.

(1) Section 2.3, page 4 – line 32: the temperature program started from 50°C for 10 seconds and was increased to 270°C at 2°C /s. You write the total run time was 180 seconds, is there a problem in the total time?

(2) The total time is correct, but after reaching 270◦C the temperature was maintained until the end of the run.

(3) Page 5, line 13: The temperature program started from 50°C (10 seconds), and was increased to 270°C at 2°C/s, and kept at 270°C until the end of the run.

(1) Section 2.3, page 5 – line 4: after every analysis are you sure that all the compounds is totally desorbed?

(2) We are absolutely sure because we run blanks after each sample, consisting of a subsequent injection of the NTME syringe.

(3) No changes has been made.

(1) Section 2.3,page 4: why didn't you use an internal standard for the quantification.

(2) The purpose of this study was to prove the potential of this technique for field measurements. However, we agree that the continuous development of this method, particularly for quantitative analysis, must include the use of a proper internal standard, which is still under research.

(3) No changes has been made.

(1) Section 3, page 5 – line 15: are the measurement of temperature and particle number concentration co- located with the sampling site?

(2) The sampling device was installed a few meters away from the measurement of

temperature and particle number concentration.

(3) Page 4, line 3: Measurements of temperature and particle number concentration were performed a few meters from the sampling system.

(1) Section 3.1, page 6 – line 3: have you tested the difference in fragmentation found between a conventional GC-MS and portable GC-MS or it's an observation from a bibliography?

(2) The mass spectra obtained by portable GC-MS were compared with the ones found in NIST 2014 spectral library search. We believe that these differences in fragmentation can be explained by the different types of detector used in the portable and conventional GC-MS.

(3) Page 7, line 1: This finding can be explained by the different types of detector used in the portable and conventional GC-MS.

(1) Figure 1, 2,3 and 4: are these the average temperature during the day or during the period of sampling ?

(2) The temperatures in all figures are average temperatures during the period of sampling, with exception to the figure 5 when the sampling was not performed.

(3) Page 6, line 11: Temperature (measured at 4.2 m height) and particle number concentration (available at http://avaa.tdata.fi/web/smart and provided by Junninen et al., 2009) were used for comparison with obtained results, and averaged during the period of sampling.

(1) Figure 3 : you quantified separately the ethylbenzene and the p/m- xylene, these two compounds that are well separated on your column ? (Tr = 0,70 min and Tr = 0,72 min). Which is the factor of resolution between these two compounds? Maybe you can considered changing your column to improve the resolution.

(2) Although the two different peaks can be clearly identified, the separation between

ethylbenzene and p/m-xylene is not idyllic, with an Rs of 1.1. For future quantitation of these compounds, a different column can result in an improvement of the resolution. However, Torion has only one column option to date. The column is integrated with the "column oven" as usually in fast GC, so changing it is highly complicated if not impossible. We asked already Torion for the different column chemistries, but no progress yet.

(3) Page 8, line 17: However, a better separation of these compounds is required for quantitative purposes, and the use of another chromatographic column should be considered for improvement of the resolution.